# SARS-CoV-2 RNA Detection on Environmental Surfaces in a University Setting of Central Italy

**DOI:** 10.3390/ijerph19095560

**Published:** 2022-05-03

**Authors:** Anna Casabianca, Chiara Orlandi, Giulia Amagliani, Mauro Magnani, Giorgio Brandi, Giuditta Fiorella Schiavano

**Affiliations:** 1Department of Biomolecular Sciences, University of Urbino Carlo Bo, 61029 Urbino, Italy; chiara.orlandi@uniurb.it (C.O.); giulia.amagliani@uniurb.it (G.A.); mauro.magnani@uniurb.it (M.M.); giorgio.brandi@uniurb.it (G.B.); 2Department of Humanities, University of Urbino Carlo Bo, 61029 Urbino, Italy; giuditta.schiavano@uniurb.it

**Keywords:** SARS-CoV-2 RNA, environmental surfaces, high-touch surface, university setting, real-time RT-PCR multiplex assay

## Abstract

The transmission of SARS-CoV-2 occurs through direct contact (person to person) and indirect contact by means of objects and surfaces contaminated by secretions from individuals with COVID-19 or asymptomatic carriers. In this study, we evaluated the presence of SARS-CoV-2 RNA on surfaces made of different materials located in university environments frequented by students and staff involved in academy activity during the fourth pandemic wave (December 2021). A total of 189 environmental samples were collected from classrooms, the library, computer room, gym and common areas and subjected to real-time PCR assay to evaluate the presence of SARS-CoV-2 RNA by amplification of the RNA-dependent RNA polymerase (RdRp) gene. All samples gave a valid result for Internal Process Control and nine (4.8%) tested very low positive for SARS-CoV-2 RNA amplification with a median Ct value of 39.44 [IQR: 37.31–42.66] (≤1 copy of viral genome). Our results show that, despite the prevention measures implemented, the presence of infected subjects cannot be excluded, as evidenced by the recovery of SARS-CoV-2 RNA from surfaces. The monitoring of environmental SARS-CoV-2 RNA could support public health prevention strategies in the academic and school world.

## 1. Introduction

The coronavirus disease 2019 (COVID-19) outbreak is a global health concern [1], with a serious impact affecting people’s lives in various aspects, including healthcare, and economic and social factors [2,3]. Since the onset of the coronavirus disease 2019 (COVID-19) pandemic, there have been over 500 million known cases of severe acute respiratory syndrome coronavirus 2 (SARS-CoV-2) infections [4]. Due to the high infectivity of SARS-CoV-2 and numerous reported cases and deaths, on 12 March 2020, the World Health Organization (WHO) declared COVID-19 as a pandemic [5]. The transmission of SARS-CoV-2 occurs through direct contact (person to person), inhalation of respiratory droplets smaller than 5 µm in diameter [6] and aerosols expelled from an infected individual during coughing/sneezing, talking or exhaling. SARS-CoV-2 can also be transmitted via indirect contact if objects and surfaces are contaminated by secretions from individuals with COVID-19 or asymptomatic carriers [7]. While aerosolized particles persist in the air for minutes to hours, exhaled droplets will settle on nearby inanimate objects and surfaces [8]. Touching these contaminated surfaces, or fomites, by an unsuspecting host can result in self-inoculation of mucous membranes of the mouth, nose or eyes. In the hospital setting, SARS-CoV-2 contamination has been detected on numerous high-contact surfaces, specifically on bed rails, tables, call panels and door handles of rooms housing COVID-19 patients [9,10]. In fact, the surfaces, as well as the hands, can represent important vehicles of contamination and potential sources of transmission of infectious agents. In the context of this pandemic scenario, the role of environment-to-human COVID-19 spread is still a matter of debate because mixed results have been reported concerning SARS-CoV-2 stability on high-touch surfaces in real-life scenarios. Up to now, very few studies have been carried out using cell culture-based systems to evaluate the infectivity of SARS-CoV-2 samples on surfaces and fomites due to limitations such as the low sensitivity and the need to have access to biosafety level 3 laboratories. Recently published data indicate that fomite transmission of SARS-CoV-2 may occur, as the virus can remain viable for days on surfaces under controlled experimental conditions, in a similar way to SARS-CoV-1 [11]. A study evaluating the duration of the viability of the virus on objects and surfaces showed that SARS-CoV-2 can be found on plastic and stainless steel for up to 2–3 days, cardboard for up to 1 day and copper for up to 4 h [12]. Knowledge about environmental contamination during outbreaks and transition phases is important to enforce public health measures intended to control viral spread from symptomatic and asymptomatic individuals [13,14]. Concerns about environmental contamination and the associated risk of indirect transmission can be raised in crowded environments (e.g., universities). Surface contamination in non-healthcare settings is still poorly studied. The presence of viral genetic material on the surfaces is not the same as the presence of infectious SARS-CoV-2 but reveals the transit and contact of infected individuals. Even if surface testing is a complement to preventive measures (disinfection programs, social distancing, employees’ protection, etc.) and environmental monitoring plans, it remains useful as part of the risk assessment to also ensure employees’ safety. It can be relevant in overcrowded environments, such as university settings, where cross-contamination between employees, students and multi-touch surfaces can easily occur through indirect transmission. For this reason, the present study was conducted during the fourth pandemic wave from 3 to 17 December 2021 inside university settings, aims to evaluate by a real-time Reverse Transcriptase Polymerase Chain Reaction (RT-PCR) multiplex assay the presence of SARS-CoV-2 RNA on surfaces and fomites in university classrooms, the library, computer room, gym and common areas that are crowded environments by working staff and students, as well as accompanying persons during graduation sessions.

## 2. Materials and Methods

### 2.1. Sampling Locations and Methods Applied

The sampling was performed from Monday to Friday between 08.00 and 09.00 a.m. (*n* = 189), before the start of teaching activities, the arrival of students or professors, and then in the afternoon at the end of the working day between 06.00 and 07.00 p.m. (*n* = 189). Standard sanitation of all university places was carried out every evening.

The environmental samples were collected in classrooms, the library, computer room, gym and common areas. As for a typical environmental monitoring program, samplings were directed to identify high-touch surfaces and fomites in university settings in indoor areas exposed to human crowding or frequently touched by hands, which included shared workstations (mouses and keyboards), computer accessories, doorknobs, tabletops, fitness equipment and vending machines. Sample collection was carried out following the protocol of the SARS-CoV-2 Surface kit (Diatheva Srl., Cartoceto, Italy). Briefly, environmental samples were collected using a swab with a synthetic tip and a plastic shaft soaked in DNAse RNAse-free water. The recommended swab surface area of 25 cm^2^ was sampled by swabbing the entire surface horizontally or vertically, rotating the swab throughout [15].

Each swab was then placed in a tube containing a guanidine solution (viral transport medium, VTM) which inactivates and stabilizes the viral genetic material (Zymo Research, Irvine, CA, USA or Zybio Inc Chongqing, China, based on availability). Since Zybio swabs are filled with 3 mL of preservation solution while Zymo has 1 mL, when using the former, 2 mL of liquid was discarded prior to collection in order to have the same amount of liquid in all samples. The samples were processed immediately or kept at +4 °C until the RNA extraction step, which was always carried out within 72 h of collection.

In order to produce artificially contaminated surfaces of different materials (plastic, metal, wood and paper), 100 µL SARS-CoV-2 RNA-containing VTM, previously obtained from nasopharyngeal swabs of confirmed SARS-CoV-2 positive patients [16], were kept in contact with the different surfaces for 15 min, then subjected to sampling and RNA extraction.

### 2.2. RNA Extraction

Total RNAs from environmental samples were extracted using a Total RNA Purification Kit (Norgen Biotek Corp., Thorold, ON, Canada) starting from 250 μL of VTM and following the manufacturer’s Supplementary Protocol for Norgen’s Saliva RNA Collection and Preservation Device. After pipetting the lysis buffer of the sample to be extracted, 0.5 µL (i.e., 1/100 of the elution volume) of a synthetic RNA process control (Internal Process Control, IPC, Diatheva Srl., Cartoceto, Italy) was added to each sample to evaluate the RNA extraction efficiency and identify the presence of PCR inhibitors. Purified RNA was stored at −80 °C until analysis.

### 2.3. Real-Time RT-PCR Multiplex Assay

Reverse Transcriptase Polymerase Chain Reactions (RT-PCR) were carried out in a 7500 real-time PCR system (Applied Biosystems, Thermo Fisher Scientific Inc., Foster City, CA, USA) using the SARS-CoV-2 Surface Kit (Diatheva Srl, Cartoceto, Italy), a molecular test designed according to WHO guidelines for the qualitative detection of SARS-CoV-2 RNA from environmental surfaces. The assay consists of a one-step real-time reverse RT-PCR multiplex assay based on fluorescently labeled probes, able to confirm the presence of SARS-CoV-2 RNA by amplification of the RNA-dependent RNA polymerase (RdRp) gene. The primers and probe for the RdRp gene are based on a previously published “discriminatory assay” specific for SARS-CoV-2 RNA detection [17], and the sequences are: Primer RdRP_SARSr-F2 5′-GTGARATGGTCATGTGTGGCGG-3′; Primer RdRP_SARSr-R1 5′-CARATGTTAAASACACTATTAGCATA-3′; Probe RdRP_SARSr-P2 FAM-CAGGTGGAACCTCATCAGGAGATGC-BBQ. The kit provides all the reagents required for PCR positive and negative controls and IPC amplification. The reaction and amplification conditions were performed according to the manufacturer’s specifications. Briefly, 5 μL of extracted RNA was added to 15 μL of the reaction mixture, and the reaction was incubated at 48 °C for 30 min and 95 °C for 10 min followed by 45 cycles at 95 °C for 15 s and 60 °C for 30 s. Fluorescence was detected during the annealing-extension step on the green channel (FAM dye) for the RdRp target and on the yellow channel (VIC/Cal Fluor orange 560 dye) for the IPC. The results were considered valid only when the cycle threshold (Ct) values of the IPC were ≤40. The results were considered positive when the Ct values of the RdRp target gene were ≥10 and negative when no amplification signal for RdRp was obtained. Invalid results (IPC undetected in RdRp negative samples) had to be re-tested.

### 2.4. Statistical Analysis

Continuous data are given as mean and standard deviation (SD) or median and interquartile range [IQR], and categorical data are given as counts and percentages. The analyses and graphs were performed using GraphPad Prism (version 8.4.2, GraphPad Software, San Diego, CA, USA).

## 3. Results

### 3.1. Validation of the Assay

Four specific matrices: plastic, metal, wood and paper, were artificially contaminated using VTM from nasopharyngeal swabs already diagnosed as SARS-CoV-2 positive to evaluate RNA recovery from the different materials and to verify the absence of PCR inhibition by the material itself. None of the four tested materials affected RNA isolation, SARS-CoV-2 RNA detection or inhibited PCR, as revealed by the positive amplification signals of both the IPC and RdRp gene (mean Ct values (SD): 32.67 (2.08) and 36.32 (2.82) for IPC and RdRp, respectively), (Appendix A, Figure A1(A.1,A.2)).

### 3.2. Detection of SARS-CoV-2 RNA on Surfaces

Between 3 and 17 December 2021, the environmental samples were collected at the University of Urbino Carlo Bo. The sampling sites were classified as low, medium and high crowding on the basis of the number of people present: between 10 and 20 (e.g., exam room), between 20 and 50 (e.g., classrooms, laboratories, common areas) and between 50 and 100 (e.g., lecture Hall, graduation room), respectively. As a control for the sanitary procedures, a subgroup of 90 samples (ten per day) collected in the morning (before any academic activities) were analyzed and tested negative for viral RNA, demonstrating the SARS-CoV-2 RNA absence in the surfaces analyzed before the entry of students and academic staff.

All samples (189/189, 100%) collected in the afternoon (after the academic activities) gave a valid result for Internal Process Control (IPC Ct ≤ 40 according to the supplier’s indications) with a median Ct value of 31.17 [IQR: 30.89–31.94], (Figure 1 and Figure A1(A.3)).

Nine samples tested positive for SARS-CoV-2 RNA (9/189, 4.8%; Figure 2) and gave a Ct value for the RdRp gene (median [IQR] 39.44 [37.31-42.66]) (Figure 1 and Table 1).

Just 1 of them had a Ct value approaching the single copy of target (Ct 34.81) while the remaining 8 were very low positive samples (according to AMCLI indications for sample Ct > 35, [18,19]): 4 had a Ct value between 35 and 40 and 4 greater than 40 (less than one copy) [20], (Figure 1 and Figure A1(A.4)). These samples were negative for the viral RNA in the paired morning test.

The PCR-negative samples were obtained from five different areas: computer station (*n* = 27), classroom desk (*n* = 87), toilet (*n* = 30) and the snack and drink vending machine (*n* = 27), gym facility (*n* = 9) and had a median IPC Ct value of 31.17 [30.89–31.96]. The positive PCR control always gave an amplification signal (mean (SD) 29.40 (1.65) and 32.52 (0.18) for IPC and RdRp genes, respectively), and the negative PCR control always gave no amplification signal for both IPC and RdRp genes, confirming the accuracy of the PCR experimental procedure. A summary table shows the number of SARS-CoV-2 RNA positive and negative samples by type of surface with the positivity rate (Table 2).

## 4. Discussion

In this study, we assessed environmental contamination with SARS-CoV-2 RNA in an academic setting frequented by students and teaching staff. Although the presence of viral RNA does not necessarily mean the presence of the infectious virus, its detection on surfaces of indoor environments could be an indicator of viral shedding from infected subjects or ineffective cleaning and disinfection. The role of contact, via contaminated surfaces, in the indirect transmission of SARS-CoV-2 is not clear, and the minimum viral load that may lead to the disease onset by contact with an infected surface is still unknown [21], just like how the contact transmission could be influenced by virus survival time on different surfaces [11,22]. The present research was carried out at the University of Urbino Carlo Bo (northern area of the Marche region, Italy) in December 2021, at the beginning of the fourth wave of COVID-19, during a significant daily increase in the number of new cases (Rt = 1.30 in Italy, [23]). Over this period, during academic lessons, many students from the various geographical areas of Italy attended university classrooms, laboratories, service halls and toilets, where indirect transmission can easily occur due to cross-contamination between employees, students and multi-touch surfaces. Although the access to university facilities, allowed only to Italian Green Pass certificate holders, wearing a face mask and with a temperature below 37.5 °C, hinders the entrance to symptomatic infected people, the environmental contamination of surfaces and objects can be ascribed to asymptomatic subjects since it has been demonstrated that these people can have a viral load similar to symptomatic ones [24,25]. In fact, the presence of asymptomatic infected subjects among vaccinated, particularly in subjects who have not yet received the third dose [26] or in subjects with a false-negative result by antigen test [27,28], cannot be excluded. In our study, 9 samples out of a total of 189 environmental surfaces sampled (4.8%) resulted positive for SARS-CoV-2 RNA. Excluding the 4 samples with <1 copy of the viral genome, the positivity rate in our study is reduced to less than 3%, and in any case, all samples were in the range of 1 copy of viral RNA. This result is in agreement with previous reports that analyzed environmental surfaces (4.26–5.25%) [29,30]. However, a direct comparison between our findings and those from similar studies is difficult since, to the best of our knowledge, there are no studies monitoring a university setting. In fact, most of the research has focused on RNA detection in hospitals and healthcare facilities [10,31,32,33], and only a few studies have explored the presence of viral RNA in non-medical environments [34,35,36]. We found positive samples from surfaces of various areas: the study allowed the identification of some critical points, such as the toilet area, the snack and drink vending machine, the handles in areas with frequent passage and big classrooms (Aula Magna) after academic activity. Other studies have also identified computer keyboards and/or mouses as at risk for SARS-CoV-2 RNA contamination [31,37]. In this investigation, the surfaces that resulted positive for SARS-CoV-2 RNA were steel, wood and plastic and the SARS-CoV-2 can remain viably infectious in these surfaces from a few hours to a few days [11]. This research has some limitations. Firstly, the presence of SARS-CoV-2 RNA in environmental samples does not necessarily indicate the presence of a viable virus. Thus, viral culturing should be performed to demonstrate viability. Furthermore, no conclusions can be reached regarding SARS-CoV-2 RNA persistence over time since the sampling was limited to a single academic setting during a limited period of 2 weeks. Moreover, repeated sampling would increase knowledge of viral RNA persistence and the effectiveness of the cleaning procedures. Indeed, all samples were collected before the disinfection operations. However, our results show that, despite the preventive measures implemented, the presence of infected subjects cannot be excluded, as evidenced by the recovery of SARS-CoV-2 RNA from surfaces. Finally, routine and extended investigations would be needed to confirm these preliminary results.

## 5. Conclusions

The containment measures adopted to avoid the introduction and spread of SARS-CoV-2 in the university environment, although efficient and adhere to the ministerial guidelines, cannot rule out the risk, most likely due to the presence of asymptomatic subjects. Although the evidence on the transmissibility of the virus through contact with contaminated surfaces is not fully understood, this possibility cannot be excluded. Hence, the rapid and efficient disinfection of indoor surfaces plays a crucial role in counteracting the contamination by the infective SARS-Cov-2 virus and should be implemented on certain surfaces and in specific periods of the academic environment. However, environmental monitoring of SARS-CoV-2 RNA could effectively support public health prevention strategies in the academic and school world.

## Figures and Tables

**Figure 1 ijerph-19-05560-f001:**
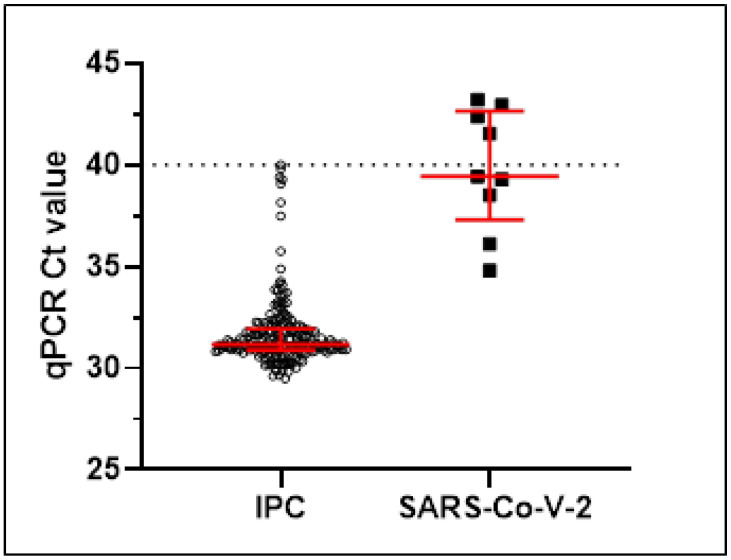
Results from qPCR in 189 environmental samples; all samples gave a valid Ct value (≤40, dotted line) for the Internal Process Control (IPC), and 9 gave a positive amplification for RdRp gene of SARS-CoV-2 RNA. Red lines represent the median and 25th to 75th percentiles.

**Figure 2 ijerph-19-05560-f002:**
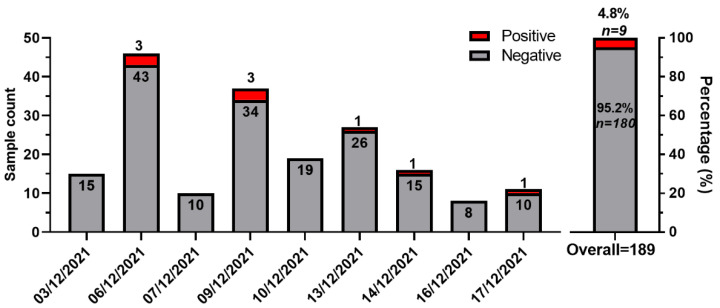
Number of environmental samples collected over 2 weeks and percentage of positive and negative samples for SARS-CoV-2 RNA.

**Table 1 ijerph-19-05560-t001:** Characteristics of positive environmental samples for SARS-CoV-2 RNA.

	ID	Date	Sampling Location	Crowding	IPC Ct	RdRp Ct
Pos_1	n 31	06/12/2021	Doorknob in toilet area	~30 students	31.03	43.21
Pos_2	n 18	06/12/2021	Snack and drink vending machine	~50 students	31.90	36.10
Pos_3	n 19	06/12/2021	Gym facility	~80 students (after activities)	31.47	41.54
Pos_4	n 58	09/12/2021	Door handle in an area with frequent passage	~60 students	32.11	39.44
Pos_5	n 66	09/12/2021	Doorknob in toilet area/flush toilet	~60 students	33.59	42.94
Pos_6	n 64	09/12/2021	Aula Magna 1	83 students (after activities)	30.77	42.37
Pos_7	n 106	13/12/2021	Aula Magna 2	~30 students	30.92	34.81
Pos_8	n 126	14/12/2021	Classroom during exams	21 students during exam and their teacher	31.87	38.52
Pos_9	n 155	17/12/2021	Door handle of the graduation room during a graduation session	~100 students and their accompanying persons and 8 teachers	31.59	39.30
Median					31.59	39.44
IQR					30.98–32.01	37.31–42.66

**Table 2 ijerph-19-05560-t002:** Number of SARS-CoV-2RNA positive and negative samples by type of surface and the positivity rate.

Type of Surface	*n*	Positive	Negative	Positivity Rate (%)
Plastic	69	3	66	4.3%
Metal	34	3	31	8.8%
Wood	60	3	57	5.0%
Paper	26	0	26	0.0%
Total	189	9	180	4.8%

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
