# Peer review of "SARS-CoV-2 RNA Detection on Environmental Surfaces in a University Setting of Central Italy"

_ijerph, 2022, doi:10.3390/ijerph19095560_

Round 1

Reviewer 1 Report

At a time when it is not fully understood how the virus spreads, it is of course welcome to investigate the persistence of the virus on surfaces around us. That is why your contribution is important.
In your manuscript, you have defined several sites where you have taken samples. At what times during the day were these samples taken; were there multiple collections at the same site at a given time interval?
It would be a good idea to have a daily sampling protocol with multiple collections at the sites (9) where the virus was detected.

Reviewer 2 Report

I have reviewed the manuscript, in which author evaluated the presence of SARS-CoV-2 RNA on surfaces made of different materials located at university environments frequented by students and staff involved in academic activity, during the fourth pandemic wave. This study showed out of 189 collected samples, 9 were tested very low positive for SARS-CoV-2 virus.

The work seems to fall in line with the journal’s scope, providing insights on the contamination by infective SARS-CoV-2 virus. All questions asked in the introduction appear to be sufficiently addressed throughout the text. Therefore, I recommend that this manuscript is ready for the publication as it is.

The main question addressed by this research: how does indirect contact by means of objects and surfaces contaminated by secretions from individuals with COVID19 do transmission of SARS-COV2?

It is relevant and interesting: The way corona virus has been spreading it is a good idea to monitor the transmission of SARS-CoV-2 via direct and indirect contacts to support public health prevention.
How original is the topic? What does it add to the subject area compared
with other published material?

Up to now, very few studies have been carried out using cell culture based systems to evaluate infectivity of SARS-CoV-2 sampled on surfaces and in real life scenario. So this is the not first study came out but very good real life based study.
The text is clear and easy to read with correct grammar.
Are the conclusions consistent with the evidence and arguments presented?

Based on 9 positive samples out of 189 collected samples, the conclusions make sense.

The main question posed in this study was that can indirect contacts spread the transmission of SARS-Cov-2 virus ? Based on 9 positive samples, this question was addressed.

Author Response

Author response:

As reported by the reviewer “I recommend that this manuscript is ready for the publication as it is.” no revision has been made and we would like to thank the reviewer for careful and thorough reading of this manuscript and for the thoughtful comments. have not made any reviews.

Reviewer 3 Report

The authors carried out a study to evaluate whether the different surfaces frequented within an Italian University showed the presence of SARS-COV-2 RNA virus.

This study was correctly carried out with the presence of the different controls necessary for the different steps of the analysis. The article is overall well documented and well written. The results provide interesting information for the knowledge of a frequented environment such as the university environment.

In my opinion, the article can be accepted when different points are added or corrected. Below you will find my different remarks according to the different parts of the manuscript.

Introduction

  • Line 34 Update the numbers (over 500M total cases currently worldwide)
  • Announcing a clear objective at the end of the introduction is missing in my opinion

Material and Methods

  • Add a summary table of the different types of samples received by surface etc.
  • Add the sequences of the different primers used for the RT-qPCR
  • When using the different acronyms for the first time, do not forget to write the whole form (for example Reverse Transciptase Quantitative Polymerase Chain Reaction)
  • Add a statistical paragraph at the end of the material and methods 2.4 (Mean value, IQR, Software used etc.)

Results

  • Rather than adding a pie chart above the chart it would be better to add an "overall" column after the last date to show the total number of positive samples and the positivity rate
  • I don’t understand why you say that 9 samples had a CT value below the threshold of 40 while on the graph of figure 3 we see that 4 of them have a value higher than 40
  • In my opinion, the PCR curves could be put in an appendix and it would be interesting to make a summary table of the number of positive and negative by type of surface with also the positivity rates. If this table is done, it is not necessary to add the one requested in materials and methods

Discussion

  • I think that the discussion could be more synthetic especially on the part concerning the presentation of the situation in Italy (Vaccination rate, type of vaccination pass, green pass etc.)
  • It would have been interesting to perform a control after the disinfections in order to verify that after that the surface was negative. Moreover, if this was not the case, the same surface could have been found positive twice since traces of viral RNA can persist over time.
